# Where Is the Oxygen? The Mirage of Non-Oxidative Glucose Consumption During Brain Activity

**DOI:** 10.3390/neurosci6040126

**Published:** 2025-12-09

**Authors:** Avital Schurr

**Affiliations:** Department of Anesthesiology and Perioperative Medicine, School of Medicine, University of Louisville, Louisville, KY 40202, USA; avital.schurr@gmail.com

**Keywords:** astrocyte, BOLD fMRI, energy metabolism, glucose, glycolysis, lactate, neuron, oxidative phosphorylation, oxygen

## Abstract

Ever since the discovery that neuronal tissue can utilize lactate as an aerobic substrate for mitochondrial adenosine triphosphate (ATP) production, a debate has ensued between those who have questioned the importance of lactate in brain energy metabolism and those who argue that lactate plays a central role in this process. The “neuron astrocyte lactate shuttle hypothesis” has sharpened this debate since it postulates lactate to be the oxidative energy substrate for activated neurons. Those who minimize lactate’s role insist that a non-oxidative process they termed “aerobic glycolysis” supports brain activation, despite oxygen availability. To explain the paradox that the active brain would utilize the inefficient glycolysis over the much more efficient mitochondrial oxidative phosphorylation (OXPHOS) for ATP production, they suggested the “efficiency tradeoff hypothesis,” where the inefficiency of the glycolytic pathway is traded for speed necessary for the information transfer of the active brain. In contrast, other studies reveal that oxidative energy metabolism is the process that supports brain activation, refuting both the “aerobic glycolysis” concept and the premise of the “efficiency tradeoff hypothesis”. These studies also shed doubts on the usefulness of the blood oxygenation dependent functional magnetic resonance imaging (BOLD fMRI) method and its signal as an appropriate tool for the estimation of brain oxygen consumption, as it is unable to detect any oxygen present in the extravascular brain tissue.

## 1. Introduction

For years, researchers studying brain energy metabolism have debated whether the glycolytic pathway or the mitochondrial OXPHOS is primarily responsible for powering the brain during activity. The year 1988 witnessed the publication of two seminal studies, each reporting results that went against the accepted dogma of brain energy metabolism. The two studies appeared in the same journal within a little more than a month apart, sparking an intense and ongoing debate among neuroscientists working in the field. The first study [1] showed that lactate, the glycolytic end-product, can support neuronal activation in vitro as the sole mitochondrial oxidative energy substrate. The second study [2] showed that glucose consumption during focal physiological neural activity is non-oxidative. The first study contradicted the accepted doctrine that lactate is a useless end-product of anaerobic glycolysis. The second study stood against the accepted, logical assumption, that neural activity requires a significant increase in ATP supply, and that the only process that could answer this increased demand is OXPHOS. As a result, the phrase “aerobic glycolysis” has been coined to refer to this occurrence; Warburg [3] originally applied it to describe how cancerous cells continue to depend on glycolysis even when oxygen is plentiful. While the phenomenon of aerobic glycolysis-supported brain activity gained acceptance as a paradigm shift in the field of brain energy metabolism, the oxidative lactate utilization to support neuronal activity was met with great skepticism. There were those who argued that its demonstration in vitro does not necessarily signify any importance in vivo. Others accepted the possibility that lactate could be an oxidative energy substrate, but only under certain limited conditions. The idea that lactate functions as an oxidative energy source for neurons gained wider acceptance among neuroscientists only six years later, upon the publication of the “astrocyte neuron lactate shuttle (ANLS) hypothesis” [4]. However, that very hypothesis also sharpened the debate between those who support the idea of lactate as the oxidative substrate for active neurons and those who support the concept of aerobic glycolysis as the major route for energy production in support of active neurons. Recently, this ongoing debate has resulted in the publication of a commentary [5], a review [6], and an editorial article [7]. The commentary article attempts to highlight the futility of the ongoing feud among the debating researchers. The review article summarizes the arguments in support of the “aerobic glycolysis” concept and against the “ANLS hypothesis”. The editorial article attempts to persuade all involved in this research field to accept a unifying concept of brain energy metabolism in rest and during activity, in health and disease. This attempt attracted two responses. The first disagreed with the major role the editorial article attributes to the ANLS hypothesis [8]. The second letter endorsed the existing debate [9]. Almost simultaneously, another review article was published [10], reviewing the studies that have led to the postulation of the ANLS hypothesis and the multitude of studies that over the past three decades helped to establish the concept of the neuron-astrocyte metabolic unit as the cornerstone of brain energy metabolism. Beyond the prolonged debate highlighted here, there is the matter of the glycolytic end-product identity, namely, lactate. It seems that too many neuroscientists, biochemists and physiologists still hold to the old division of this pathway into aerobic and anaerobic ones, where the former ends up with pyruvate, and the latter ends up with lactate. Yet, paradoxically, they accept that “aerobic glycolysis” provides the necessary ATP for the active brain, where lactate is its end-product (anaerobic glycolysis). Clearly, holding on to this old division is not just confusing, but continuing to promote the dogmatic and wrong concept that normally, under aerobic conditions, the glycolytic end-product is pyruvate [11,12,13]. Hu and Wilson [14] published a significant study in 1997, though proponents of the “aerobic glycolysis” theory often overlook it. The research showed that neurons consume oxygen upon activation. Two recent studies, one by Morelli and Scholkmann [15], and the other by Vervust et al. [16] appear to re-enforce the findings of Hu and Wilson [14], showing that brain tissue, especially its myelin, can store ample amount of oxygen, making it available during brain activation, and therefore not relying on blood-borne oxygen. These studies also intensify the already accumulating doubts about the usefulness of the BOLD fMRI signal as a reliable tool for brain oxygen consumption measurements [5,11,12,13].

## 2. A Brief History of the Active Anaerobic Brain

“Non-oxidative glucose consumption during focal physiologic neural activity,” the title of the article by Fox et al. [2], challenged the conventional view of brain energy metabolism. The increased utilization of glucose without an accompanied increase in oxygen consumption upon brain stimulation meant that the brain resorts to glycolysis rather than oxidative energy production to support its energy needs during activity. This unexpected observation received support when Prichard et al. [17] published their findings using ^1^H NMR to detect lactate rise in the human visual cortex during physiological stimulation. The abstract of their publication reads: “Brain lactate concentration is usually assumed to be stable except when pathologic conditions cause a mismatch between glycolysis and respiration. Using newly developed ^1^H NMR spectroscopic techniques that allow measurement of lactate in vivo, we detected lactate elevations of 0.3–0.9 mM in human visual cortex during physiologic photic stimulation. The maximum rise appeared in the first few minutes; thereafter lactate concentration declined while stimulation continued. The results are consistent with a transient excess of glycolysis over respiration in the visual cortex, occurring as a normal response to stimulation in the physiologic range”.

These findings, especially originating from the renowned laboratory of Prof. Shulman at Yale University, helped establish the idea that non-oxidative glucose utilization energizes neural activity. In 1991, Raichle published a book chapter [18] where he conclusively determined “that phasic neural activity is supported by glycolysis,” a determination that prompted him to ask several questions and to provide answers. Here are two relevant questions to this monograph and the answers offered:

“Why glycolysis?” Because “…the use of glycolysis to support abrupt changes in local neuronal activity make it unlikely, over short times, that the tissue energy demands will outstrip energy supply. This occurs because the tissue concentrations of glucose and glycogen are sufficient to sustain neural activity over several minutes. This should be contrasted to oxygen which is at near zero concentration in the tissue”.

“Is glycolysis enough?” Raichle used an estimation of “…about 0.3 to 3.0%, or even less, of the cortical energy consumption can be accounted for by spike activity of cortical nerve cells”. His answer: “A doubling of neural electrical activity, therefore, should increase oxygen consumption by less than 6%, in agreement with the small changes that we have observed”. Since “glucose oxidation is at (or near) maximal capacity at rest and that increases of glucose oxidation during stimulation, whether natural or experimental, cannot therefore take place, unless by a few percent”.

The answer to the first question assumed a near zero concentration of oxygen in the tissue outside the vasculature. The answer to the second question assumed that glucose oxidation at rest is already at or near maximum, such that any significant increase in glucose oxidation upon stimulation, considering the near zero tissue oxygen concentration, is minimal at best. In summary, neurons need little oxygen for spiking activity because this process only requires a small amount of ATP, which glycolysis can easily provide. Additionally, since glucose oxidation at rest is already close to maximum, a significant increase in glucose oxidation to support spiking activity is unlikely. However, it is unclear if this conclusion is based on the determination that tissue oxygen is close to zero. Both the assumptions and the answers to the questions rely on the measurements performed in Prof Raichle’s laboratory, where upon neural activation the increase in glucose consumption was unaccompanied by oxygen consumption.

As awareness of non-oxidative glucose consumption (“aerobic glycolysis”) increased, researchers began utilizing lactate level measurements to evaluate metabolic activity after brain stimulation [19,20]. By 1994, Pellerin and Magistretti published their hypothesis paper [4], describing the uptake of the excitatory neurotransmitter, glutamate, by astrocytes upon its release from neuronal synapses. This process involves sodium ions, which stimulate the astrocytic Na^+^/K^+^-ATPase pump, powered by glycolysis. Additionally, they proposed that the lactate generated during that pumping process is subsequently transported back to neurons that use it for energy production through mitochondrial OXPHOS. The proposed hypothesis laid out a metabolic map that could explain the phenomenon observed by Fox et al. [2] of brain activity supported by non-oxidative glucose consumption (astrocytic sodium ions pumping), but it also proposed oxidative lactate consumption, the oxygen for which the study by Fox et al. [2] did not detect.

Importantly, all the studies that support the concept of ‘brain activation by non-oxidative glucose utilization estimated oxygen consumption during activation by measuring either changes in the rate of cerebral blood flow (CBF) or changes in the level of blood hemoglobin oxygenation (deoxyhemoglobin concentration) or both. Neither measured changes in O_2_ levels of brain tissue directly nor CO_2_ production.

## 3. A Brief History of the Active Aerobic Brain

Respiration is the main and most efficient way mammals generate energy. It is well known that both glucose and oxygen are crucial for respiration, especially in the brain, which uses more of these resources by weight than any other organ. One hundred years ago, scientists discovered that oxygen exposure could reduce lactate levels in brain tissue, well before the function of mitochondria in energy metabolism was known [21,22,23,24,25,26,27]. Further research from the same lab showed that brain stimulation leads to lactate production. These studies also suggested that lactate could originate from other internal sources within the brain and that its oxidation may contribute to brain metabolism [27]. These studies received little attention since their results challenged the prevailing belief that lactate was simply a waste product of glycolysis and where glucose is the only oxidizable fuel. Six decades passed before these studies were rediscovered [28].

S. S. Kety developed the original methods to measure CBF and distribution of gases in the brain. They were based on the idea that the brain absorbs oxygen through the arterial blood, where its accumulation depends only on its diffusion, solubility, and perfusion, not on any metabolic activity [29]. Kety’s chapter, “The general metabolism of the brain in vivo” in the book “Metabolism of the nervous system”, outlined core brain energy metabolism parameters that remain widely used today [30]. For instance, the respiratory quotient of the brain (O_2_ consumed/CO_2_ produced) should be ~1.0; the cerebral metabolic rate of oxygen over that of glucose (CMRO_2_/CMRglu) supposes to approach 6.0, and only an insignificant consumption of any other substance except glucose by the normal brain takes place. While measurement techniques of both glucose and oxygen consumption have improved and new ones have developed, the basic concepts regarding brain energy metabolism have not changed until the publication of the study by Fox et al. [2]. Nevertheless, the new techniques have offered the opportunity to simultaneously measure the consumption of both glucose and oxygen and introduced the possibility of performing such measurements both at rest and during brain stimulation. The most popular technique today, BOLD fMRI, appears to lend support to the concept that non-oxidative glucose consumption fuels brain activation.

## 4. The Brain on Oxygen

According to Kety [30] “The brain appears to derive its energy, at least in the normal state, almost entirely from aerobic processes. It utilizes oxygen at a rate which is among the highest in the body, having a mean value in normal man close to 3.5 mL per hundred grams of brain per minute. For a human brain of average weight, this corresponds to a total oxygen consumption of approximately 50 mL per minute or about 20% of the basal oxygen requirements for the body as a whole”. Traditionally, estimation of oxygen consumption is determined by subtracting venal oxygen concentration from its arterial concentration. Using measurements in humans performed by various investigators under several conditions (“epileptics, schizophrenics, psychotics, hospital patients, normal young adults and normal adults”), Kety calculated the oxygen/glucose utilization ratio of the human brain to be 5.54, a value different from the theoretical value of 6.0 (6O_2_ + C_6_H_12_O_6_ —› 6CO_2_ + 6H_2_O). He postulated that such a difference is due to other metabolic processes that utilize glucose without oxidation. Nevertheless, in the 1950s, oxygen consumption estimations relied on measurements of arteriovenous differences in oxygen concentration and on measurements of CBF via the nitrous oxide method [29]. Although other methods to measure CBF were invented, including those that use radioactive tracers, they are all based on the same core idea: evaluating CBF to determine the brain’s oxygen consumption. While direct measurements of tissue oxygen concentration were available, such as the polarographic method which has been employed successfully in vitro, they were not applicable for use in humans. Consequently, when studying brain energy metabolism in vivo both in humans and other animals, measuring oxygen consumption steadfastly relies on the methods and techniques developed to measure CBF and vascular oxygen concentration to estimate oxygen consumption. The BOLD fMRI technique [31], which has become the mainstay of neurocognitive research, is based on changes in CBF and the level of blood oxygenation when investigating brain energy metabolism during stimulation. The BOLD signal amplitude and its detection depend on a change in the ratio of oxyhemoglobin to deoxyhemoglobin concentrations. Countless studies over the past three decades have relied on this method to investigate the cerebral energy metabolic demands during rest and activity. The opportunity to monitor and correlate brain activity with the metabolic processes that support it has been the great promise of this methodology. Among the many studies that use BOLD fMRI, both a linear correlation between glucose and oxygen consumption upon brain stimulation [32,33] and uncoupling between the two have been shown [34]. There are additional drawbacks when using the BOLD signal as a reliable quantitative estimate of oxygen consumption. A study that attempted to correlate the BOLD signal with direct polarographic oxygen measurements found a mismatch between the two [35]. Another study concluded that due to strong sensitivity of fractional changes in CBF to cerebral metabolic rate of oxygen consumption (CMRO_2_), the amplitude of the BOLD response cannot be interpreted as a quantitative reflection of underlying metabolic changes [36]. A more recent study indicates that “a meaningful interpretation of stimulus-induced BOLD responses should consider slowly developing variations in baseline BOLD signals and therefore, baseline correction tools should be cautiously used for fMRI data analysis” [37]. Several models [38,39,40] endeavor to explain aerobic glycolysis, where the brain meets increased energy demands through the less efficient glycolytic pathway. Accepting the use of CBF measurement and BOLD signal as the best methods available to estimate brain oxygen consumption also meant accepting the non-oxidative glucose consumption paradox. Therefore, Theriault et al. [41] offered “the efficiency tradeoff hypothesis,” according to which the active brain chooses speed over efficiency when energy (ATP) is in high demand. At about the same time Baxton and Frank [38], Gjedde [39] and Hyder et al. [40] offered their models to describe oxygen diffusion from blood into brain tissue, Hu and Wilson [14] published their study, which employed microsensors to simultaneously track changes in glucose, oxygen and lactate in a rat brain in vivo during both rest and activity (Figure 1). In that study, they traced the dynamic changes in the levels of these three energy substrates upon stimulating the hippocampal perforant path and recording the changes induced by the stimulation in the dentate gyrus. The findings of this study clearly showed that upon activation the brain utilizes all three substrates. More importantly, oxygen is abundantly available and consumed in the oxidation of both glucose and lactate. The investigators applied ten consecutive stimulations, two minutes apart, and with each stimulation they recorded oxygen consumption. Concomitantly, they also recorded glucose and lactate consumption. While glucose levels dropped below their baseline, lactate, except immediately following the first stimulation, increased by 60% and by 100% through the eighth to tenth stimulations. As the stimulation progressed, glucose consumption declined, while lactate consumption increased. Furthermore, during the 25 min post-stimulation period, lactate became the main energy source, sparing glucose, as Figure 1 clearly indicates with an increase in glucose tissue levels above baseline, while lactate tissue levels declined. Not surprisingly, those who subscribe to the non-oxidative glucose utilization as the process that provide ATP to the active brain, find reasons to reject these findings, as reflected by the following quote: “…Hu and Wilson studies were based on incremental changes in glucose and lactate levels from baseline during electrical stimulation; absolute values were not reported. Relative concentration changes provide useful information, but they do not reflect pathway fluxes or substrate utilization rates because concentration is the net result of input to and output from the pool.” [5]. However, the baseline level of glucose in a rat brain is 2–3 µmol/g as documented in Dienel’s own publication [42]; lactate brain level fluctuates between 1 µmole/g and 4 µmole/g. As a result, it is possible to determine incremental changes from the baseline and convert them into absolute values. Hu and Wilson [14] determined the brain glucose baseline concentration (2.6–2.9 mM). The initial rise in brain lactate levels after the first stimulation (58%) can help determine the baseline concentration of lactate, while considering that each mole of metabolized glucose produces two moles of lactate. Since the amount of glucose consumed upon the first stimulation was 0.435 mM (the first glucose dip upon stimulation), the amount of lactate produced glycolytically should be 0.87 mM. As a result, the baseline concentration of lactate was determined to be 1.39 mM. Glucose and lactate, as well as oxygen, were consumed, as all three traces dipped right after each stimulation. However, while the baseline level of glucose seemed to drop by 20% after three stimulations, the lactate level rose by 58% immediately following the first stimulation and by 100% following the last three stimulations. On the other hand, oxygen levels stayed stable throughout the experiment, showing only small dips immediately upon each stimulation, signaling that there was no shortage of oxygen to be consumed in the oxidation of lactate as evidenced by the dips in the latter level following each stimulation (Figure 1).

Relying on Hu and Wilson’s own measurements of baseline glucose concentration (2.9 mM) [14] and on the calculated baseline lactate concentration (1.39 mM), the dips in the level of these two substrates were measured here and the amount of both glucose and lactate utilized during each of the ten stimuli delivered to the rat hippocampal dentate gyrus were calculated using these measurements (Figure 2).

The highest glucose consumption occurred following the first stimulation, combined with the lowest lactate consumption. Afterwards, glucose consumption was reduced by almost 50% throughout the experimental period, while lactate consumption steadily increased to over 250% by the 9th and 10th stimuli. Obviously, as the consumption of lactate increased so did the consumption of oxygen. Lactate could only be consumed via mitochondrial oxidation, which means that for a single mole of lactate three moles of oxygen are being utilized (C_3_H_6_O_3_ + 3O_2_ —› 3CO_2_ + 3H_2_O). Accordingly, the oxygen consumption values shown in Figure 2 are three times the values of lactate consumption. The results of Hu and Wilson [14] illustrate and confirm the ANLS hypothesis [4], according to which most of the glucose is consumed by astrocytic glycolysis that is responsible for the significant increase in lactate levels. Consequently, lactate is transported to neurons, where it is being utilized via mitochondrial OXPHOS, which consumes three moles of oxygen for every mole of lactate. Once brain activation ceased, lactate slowly decreased towards its baseline level, while glucose increased above its baseline level as can be seen in Figure 1. These changes indicate that during rest, when both glucose and lactate are available, lactate becomes the preferred substrate over glucose, sparing the latter, and therefore the observed increased in glucose level above baseline. One can also surmise that a significant portion of the accumulated lactate during and following activation is transported out of the brain via the venal vasculature.

## 5. To Breathe or Not to Breathe?

It is therefore clear that the determination by Dienel et al. [5] that “…Hu and Wilson studies were based on incremental changes in glucose and lactate levels from baseline during electrical stimulation; absolute values were not reported. Relative concentration changes provide useful information, but they do not reflect pathway fluxes or substrate utilization rates…” is not accurate. Moreover, according to Dienel [42] “…rat brain glucose level is ∼20% of that in arterial plasma and ranges from ~2 to 3 μmol/g. If CMR_glc_ abruptly doubles and rises from 1 to 2 μmol/g/min, then 1 μmol/g would be consumed in one min. This leaves 1–2 μmol/g in the brain precursor pool of unmetabolized glucose, even without the continuous supply of glucose from blood at a rate that matches the resting CMR_glc_. Rapid blood flow upregulation and excess delivery of oxygen and glucose indicate that normal brain does not ‘need’ lactate as a supplemental fuel. Some extracellular lactate is probably consumed at stimulus onset, but analysis of available data indicates that this lactate makes a minor contribution to CMR_glc_”. The argument by Dienel et al. [5] aligned with Raichle’s determination [18]. More recently, a non-oxidative (aerobic glycolysis) glucose consumption has been suggested to be the main energy source for activated brain, since measurements of oxygen delivery using BOLD fMRI signal and CBF indicate that no such delivery occurred [41]. Evidently, these two opposing views are offered by the same group of investigators: On one hand, is the view that the brain has enough glucose and oxygen to support all its energy needs during activation, such that “lactate is not needed as a supplemental fuel” [42], while on the other hand, they hypothesize that oxygen does not participate in the process that provides the energy for activated brain [41]. As mentioned earlier, the conclusion that oxygen does not participate in fueling the brain activity is based on estimating oxygen utilization by measuring changes in its blood level (BOLD) or changes in CBF or both. However, if oxygen stores exist within the brain tissue outside the vasculature, as the study of Hu and Wilson [14] suggests, and if these oxygen stores are readily available and more than adequate to support oxidative energy metabolism, one should not expect to observe significant changes in the BOLD signal; this lack of changes could lead one to wrongly conclude that brain activation is not accompanied by oxygen utilization. Recently, Morelli and Scholkmann [15] suggested that the myelin sheath could be an efficient site for O_2_ storge, so much so that it operates like an O_2_-sponge. Vervust et al. [16] used a diffusive model derived from molecular dynamics simulations to demonstrate that oxygen storage in a bilayer follows first-order kinetics. For myelin, having multiple bilayers, they modeled oxygen transport through a ladder network of resistor-capacitor-like circuits, where oxygen permeation resistance and oxygen storage capacity scale linearly with bilayer count. According to their model, myelin sheaths are compact oxygen reservoirs. Therefore, these investigators suggest that during increased neuronal activity, myelination prolongs the brain’s ability to sustain increased oxygen demand, preventing oxygen fluctuations, as observed by Hu and Wilson [14] (see also Figure 1).

A paradox presented by Fox et al. [2], widely accepted by researchers studying brain energy metabolism, suggests that non-oxidative glucose consumption drives brain activation. While all agree it to be a paradox, the reliance of most investigators in the field on measuring brain oxygen consumption by indirect techniques such as BOLD fMRI and CBF, prevented them from questioning the validity of the paradox, since their measurements confirm it. But these techniques assume that all the oxygen consumed by the brain, whether during rest or during activity, is in the vasculature, tied to hemoglobin. Accepting such an assumption to be true, the non-oxidative glucose consumption by the active brain seems to be a valid paradox that requires logical explanation. “The efficiency tradeoff hypothesis” [41] offers such an explanation. It suggests that the meager glycolytic ATP production is sufficient to support the main facet of brain activity, namely, informational transfer that takes place in thin, mitochondria-less axons, where the brain sacrifices efficiency for speed. That is, since producing ATP glycolytically is faster than producing it mitochondrially via OXPHOS. It allowed its advocates to conclude that brain activation requires only a minute increase in energy supply over the amount required during rest, contradicting the common knowledge according to which brain activity demands a significant increase in energy supply. Where axons are concerned, they are not devoid of mitochondria. A study by Ohno et al. [43] shows a juxtaparanodal/internodal enrichment of stationary mitochondria and neuronal activity-dependent dynamic modulation of mitochondrial distribution and transport in nodal axoplasm. Studies by Perge et al. [44] and by Giacci et al. [45] clearly shows that there are axonal elongated mitochondria that can reach the length of 7–8 µm and a width of only a fraction of 1 µm. Considering these studies along with those of Morelli and Scholkmann [15] and Vervust et al. [16] that demonstrated the ability of axonal myelin to store oxygen, the validity of “the efficiency tradeoff hypothesis” [41] is disputable. Consequently, when a published study clearly shows that oxygen is both present and consumed upon brain activation [14], those who support the paradox of non-oxidative glucose consumption (aerobic glycolysis) either ignore that study or point out its presumed shortcomings [5]. Two recently published studies strongly support the principal arguments raised in the present monograph. The first study employed a model of human brain energy metabolism using Computational Singular Perturbation [46]. The analysis performed using this model indicates that fast (seconds) energy changes linked to neuronal activity are too fast and brief to be detected by most imaging methods, those that are captured are slower changes, such as the ones that probably occur in astrocytes (non-oxidative glucose consumption to support the Na^+^/K^+^-ATPase pump). Additionally, the analysis results suggest that both glucose and lactate function as primary fuels for the brain. These conclusions agree with Hu and Wilson [14], who recorded traces of both the faster (seconds) and the slower (minutes) responses and demonstrated the utilization of both glucose (by astrocytes?) and lactate (by neurons?) non oxidatively and oxidatively, respectively (Figure 1). In vitro studies using cell cultures and brain slices add support to these conclusions, as they demonstrate utilization of lactate for neuronal function, glycolytic glucose utilization by astrocytes and lactate transport to neurons [47,48,49,50,51,52,53]. Several in vivo studies also show lactate utilization by the human brain by injecting hyperpolarized ^13^C-lactate, which is immediately converted to ^13^C-pyruvate and oxidized, a conversion that occurs most likely by mitochondrial lactate dehydrogenase [54]. Moreover, lactate appears to be attracted to the brain’s gray matter more than to its white matter [55]. The analysis by Patsatzis et al. [46] also supports the ANLS concept. In another study, researchers evaluated human volunteers to understand how their brains managed elevated lactate levels, generated either by intravenous lactate infusion or by performing vigorous interval exercise [56]. The findings of this study show that when circulating lactate is made available, whether through infusion or intense exercise, the human brain prefers metabolizing lactate over glucose for its energy needs, while sparing glucose, which could be utilized for other, non-energetic purposes in the brain. In contrast, several recent studies either support the non-oxidative glucose consumption as the metabolic process that provide ATP to active neurons, or that lactate is not the oxidative mitochondrial substrate of OXPHO. An in vitro study determined that lactate transport from astrocytes via monocarboxylate transporter 1 (MCT1) to neurons via MCT2 is not necessary to maintain synchronized synaptic transmission [57], and a recent editorial highlights these findings to argue for “the absolute necessity of neuronal glucose metabolism to maintain brain function.” [58]. Another study shows that transport of pyruvate by the mitochondrial pyruvate carrier (MPC) regulates presynaptic metabolism and neurotransmission [59]. This finding does not necessarily indicate that lactate is not the final glycolytic end-product and the oxidative mitochondrial substrate. The MPC is located on the inner mitochondrial membrane [60]. However, the mitochondrial MCT is located on the outer membrane and is responsible for lactate transport into the mitochondrion, where it is oxidized by mitochondrial lactate dehydrogenase (mLDH) to pyruvate, which is then transported by the MPC to be converted to acetyl Co-enzyme A (acetyl CoA) [61,62,63]. Wei et al. [64] suggested a compartmentalization of neuronal energy metabolism, where OXPHOS takes place mainly in axonal terminals, while glycolysis is the main supplier of ATP in neuronal somata. The investigators suggest that their findings support the concept of non-oxidative glucose consumption as the principal metabolic ATP producer during neuronal activity. They also found that the glycolytic enzyme pyruvate kinase 2 (PKM2) is more prevalent in the neuronal somata than in the terminals. Deletion of the gene *Pkm2* in mice caused neuronal somata to switch from glycolysis to OXPHOS, which increased oxidative damage. The role of glycolysis in protecting neurons against oxidative damage has been documented [51], where the protection is provided by the glycolytic end-product, lactate, not by pyruvate. Therefore, it is worth re-emphasizing that the glycolytic pathway always ends with lactate, the oxidative mitochondrial substrate, and an effective neuroprotectant against oxidative damage. As investigators continue to promote the dogma that glycolysis ends with pyruvate (in the presence of oxygen), the needless debate over which metabolic process fuels the active brain will continue to divide the research field of brain energy metabolism. Finally, since the BOLD fMRI technique has become the leading methodology in a multitude of studies investigating cerebral energy metabolism in rest and during activity, in health and disease, it is important to consider not only its promises, but some of its pitfalls and drawbacks [65,66,67]. To this list of the drawbacks, one should add its inability to detect extravascular oxygen, oxygen that active neural tissue utilizes for mitochondrial ATP production.

Figure 3 depicts two major concepts that divide the brain energy research community: The non-oxidative glucose consumption as the main metabolic process to support the active brain (left), and the oxidative astrocytic lactate consumption to support the active brain (right).

## 6. Conclusions

The concept that non-oxidative energy metabolism (aerobic glycolysis) fuels brain activation has captivated the attention of biochemists, physiologists, and neuroscientists for over three decades, despite it being a paradox. Its fundamentals have stood against everything that science already established about the energy needs of the brain, the one organ that consumes more energy per weight than any other. Consequently, it created a rift in the brain energy metabolism research community. To postulate that the active brain’s high energy needs are somehow answered by an inefficient energy producing process is, by itself, paradoxical, a mirage that only complicates and delays our understanding of the energy metabolic processes of the active brain. It is important to emphasize that BOLD fMRI, the one technique used today almost universally to measure brain oxygen consumption, and which provides most of the support for this concept, could be at best unreliable and maybe even misleading, as it only estimates blood oxygen content, not any oxygen content outside the vasculature. Additionally, the use of the term “aerobic glycolysis” is confusing and can be misleading [5,11,12,13]. The glycolytic pathway, through 11 enzymatic steps, hydrolyzes one mole of glucose to produce two moles of lactate, two moles of ATP and two moles of NAD^+^, independent of the presence or absence of oxygen. In summary, (a) the “aerobic” or “anaerobic” prefix attached to the term “glycolysis” is meaningless; (b) the mitochondrial tricarboxylic acid (TCA) cycle and its coupled OXPHOS is the most efficient energy producing pathway in the brain and elsewhere; (c) neurons prefer glycolytic lactate over glucose as their oxidative mitochondrial substrate when the concentration of the monocarboxylate rise above its resting level. The inability of the BOLD fMRI technique to detect extravascular oxygen puts doubt on its usefulness when studying the energy metabolism of the active brain in vivo.

## Figures and Tables

**Figure 1 neurosci-06-00126-f001:**
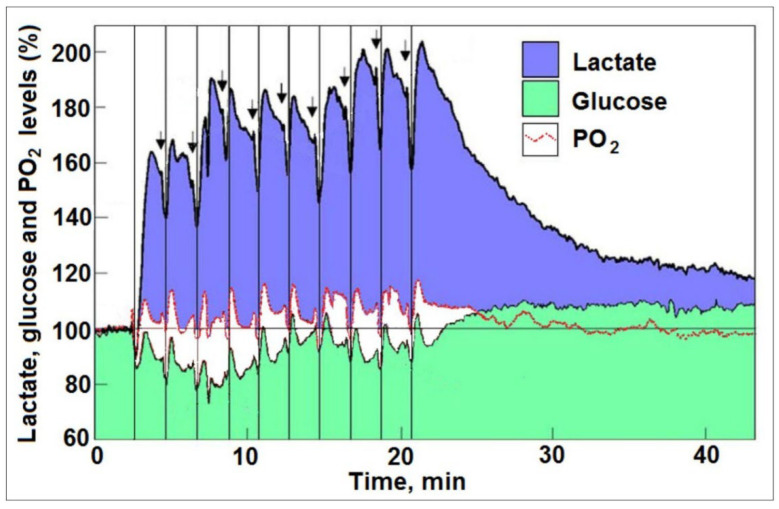
Recorded levels of local extracellular lactate, glucose, and PO_2_ levels in the rat hippocampal dentate gyrus over time during a series of 5 s electrical stimulations (arrows) of the perforant pathway at 2 min rest intervals (reproduced with permission from [14], copyright, Blackwell, Oxford). The changes in the mean concentration of glucose were always in opposite direction to the changes in mean lactate concentration except following the first stimulation. The vertical lines drawn to indicate the simultaneous dip in all three analytes in response to each of the electrical stimulations. While glucose level dropped below the baseline through the first six stimuli, lactate level increased immediately following the first stimulus and stayed as high as double the baseline throughout the stimulation period. Oxygen level remained steady throughout the experimental period exhibiting only small fluctuations (dips) immediately upon each stimulation followed by a corresponding overshoot. These fluctuations, unlike those recorded for glucose and lactate, cannot lend themselves to determination of the amount of oxygen used during each stimulation. However, oxygen consumption during each stimulation could be estimated from the dip in lactate level.

**Figure 2 neurosci-06-00126-f002:**
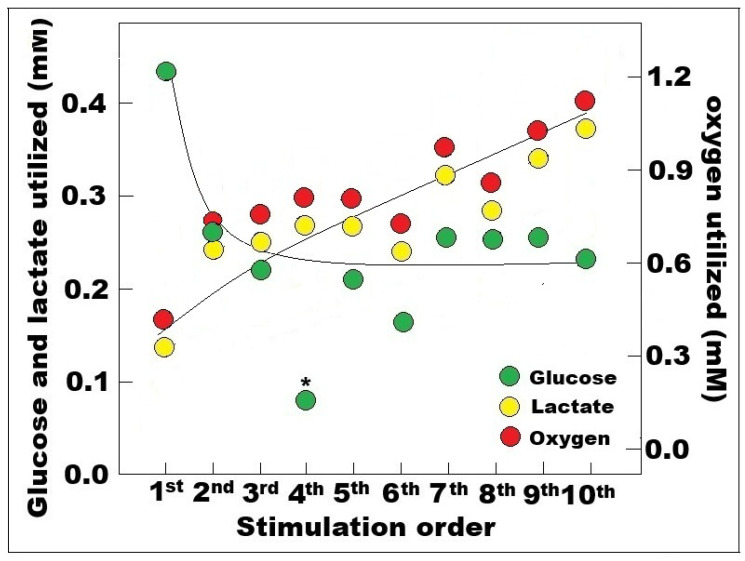
Glucose, lactate, and oxygen consumed upon each of the ten stimulations of the rat hippocampal perforant path and their responses in the dentate gyrus as recorded by Hu and Wilson [14] (see Figure 1). Glucose consumption was determined by measuring the decrease in tissue glucose levels after each stimulation, using a baseline value of 2.9 mM. Estimations of lactate consumption were achieved by measuring the dips in lactate concentration trace following each stimulation, beginning from an initial level of 1.39 mM (see the main text for more information). Oxygen consumption was determined by taking the lactate consumption measurements and multiplying them by three. * The dip in the glucose level value following the fourth stimulation was an outlier for unknown reason.

**Figure 3 neurosci-06-00126-f003:**
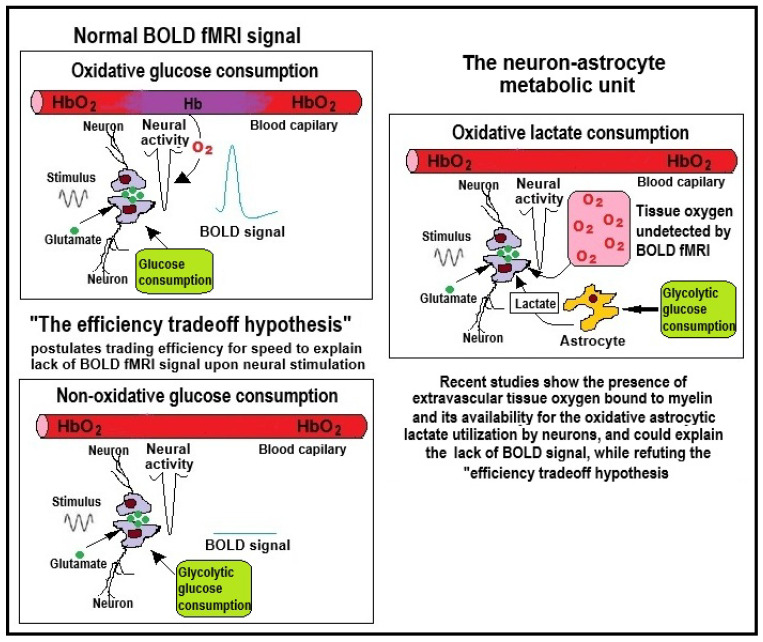
An illustration of the two opposing concepts dividing the brain energy metabolism research community. Accordingly, one group relies on studies using CBF and BOLD fMRI signal, profess that the energy metabolic pathway fueling neural activity is a non-oxidative one, and put forward “the efficiency tradeoff hypothesis” [42] (left side). In contrast, the other group claim that the energetic process supporting neural activity is the oxidative consumption of astrocytic lactate, promoting the “astrocyte neuron lactate shuttle” hypothesis [4] (right side).

## Data Availability

No new data were created or analyzed in this study. Data sharing is not applicable to this article.

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
