# Peer review of "Where Is the Oxygen? The Mirage of Non-Oxidative Glucose Consumption During Brain Activity"

_neurosci, 2025, doi:10.3390/neurosci6040126_

Round 1
Reviewer 1 Report
Comments and Suggestions for Authors
The manuscript "neurosci-3944669" by Avital Schurr is a very nice short review, which summarizes concepts on the role of oxygen and lactate utilization in the brain. The idea of this manuscript is appealing and certainly worth writing.
The abstract needs a better concluding sentence. Alternatively, if the Author suggests the sentence about usefulness of BOLD fMRI as the main concluding one, it should be made clearer from the abstract.
Example references are required for the sentence "Among the multitude of studies that used BOLD fMRI, there are those that showed linear correlation between glucose and oxygen consumption upon brain stimulation, while others indicated uncoupling between the two, a nonoxidative glucose consumption upon brain activation..."
The text of page 9 (please add lines for Reviewers) starting from "These conclusions agree with Hu and Wilson [8]," and further, is of particular interest. Additional sentences or a paragraph describing the use of lactate by cultured cells is required. Of course, such cells are a completely different system compared to the whole brain, but the ability of neurons to use lactate, especially the sentence "the human brain prefers metabolizing lactate over glucose for its energy needs" requires such analysis and discussion. Neurons and astrocytes would suit this text the best.
Minor comment: "one µmole/g and four" (page 5) should be unified with the "level of glucose in rat brain is 2-3 µmol/g" one line above.
Author Response
My thanks to Reviewer 1's comments and suggestions that greatly improved the manuscript.
The concluding sentence is now included in the abstract: 'These studies also shed doubts on the usefulness of the BOLD fMRI method and its signal as an appropriate tool for the estimation of brain oxygen consumption, as it is unable to detect any oxygen present in the brain tissue outside the vasculature."
I agree with the reviewer's point that example references are required to support the sentence "Among the multitude of studies that used BOLD fMRI, there are those that showed linear correlation between glucose and oxygen consumption upon brain stimulation, while others indicated uncoupling between the two, a nonoxidative glucose consumption upon brain activation..." Hence, refs. 32-37 were added.
The reviewer suggested to expand here: "The text of page 9 starting from "These conclusions agree with Hu and Wilson [8]," and further, is of particular interest. Additional sentences or a paragraph describing the use of lactate by cultured cells is required. Of course, such cells are a completely different system compared to the whole brain, but the ability of neurons to use lactate, especially the sentence "the human brain prefers metabolizing lactate over glucose for its energy needs" requires such analysis and discussion. Neurons and astrocytes would suit this text the best."
Therefore, the following text and references were added: "In vitro studies using cell cultures and brain slices add support to these conclusions, as they demonstrate utilization of lactate for neuronal function, glycolytic glucose utilization by astrocytes and lactate transport to neurons [47-53]. Several in vivo studies also show lactate utilization by the human brain by injecting hyperpolarized 13C-lactate, which is immediately converted to 13C-pyruvate and oxidized, a conversion that occurs most likely by mitochondrial lactate dehydrogenase [54]. Moreover, lactate appears to be attracted to the brain’s gray matter more than to its white matter [55]."
Minor comment: "one µmole/g and four" (page 5) should be unified with the "level of glucose in rat brain is 2-3 µmol/g" one line above. This has been corrected.
Reviewer 2 Report
Comments and Suggestions for Authors
In this review manuscript, Schurr questions the relevance of non-oxidative glucose consumption during brain activity, providing a concise historical perspective and casting doubts on the validity of BOLD fMRI. The manuscript is strong, both embracing and respectfully engaging in the scientific debate around this issue. The following are recommendations for improving the manuscript.
While presenting important and timely discussion regarding the oxidative/non-oxidative glucose disposal issue in brain, the manuscript in some areas reads more like a commentary on Dienel et al’s recent review (PMID: 40476345); and would therefore benefit by explicitly contextualizing its contribution among other commentaries (PMIDs: 40686252, 40966093, 40810250) and/or distinguishing its scope and depth. For example, the manuscript focuses heavily on oxygen while ignoring additional evidence for facets of the ANLS hypothesis, such as relatively recent hyperpolarized-13C MRI research (PMIDs: 38230992, 31557546) with regard to non-oxidative and oxidative sources of CO2 in astrocytes vs neurons.
The manuscript would also serve as a stronger review - especially for a broader readership - if it addressed the many shortcomings of the BOLD methodology in greater detail.
Likewise, expanding the end of section 5 to further review evidence (e.g., ref 39) in which elevated blood lactate - either endogenously or by infusion - demonstrates cerebral lactate disposal, and including the evidence for oxidative and non-oxidative metabolism, would strengthen the manuscript.
Additional comment on recent related research (e.g., PMID: 41048117, 41065327) would also strengthen the manuscript.
The chemical equation in the last paragraph of section 4 should be balanced.
Reference 39 does not appear to have been cited in the text.
Author Response
I thank Reviewer 2's comments and suggestions.
While presenting important and timely discussion regarding the oxidative/non-oxidative glucose disposal issue in brain, the manuscript in some areas reads more like a commentary on Dienel et al’s recent review (PMID: 40476345); and would therefore benefit by explicitly contextualizing its contribution among other commentaries (PMIDs: 40686252, 40966093, 40810250) and/or distinguishing its scope and depth. For example, the manuscript focuses heavily on oxygen while ignoring additional evidence for facets of the ANLS hypothesis, such as relatively recent hyperpolarized-13C MRI research (PMIDs: 38230992, 31557546) with regard to non-oxidative and oxidative sources of CO2 in astrocytes vs neurons.
Yes, there is a long-standing dialogue between Gerry Dienel and me, some of which touched on by Dienel in his review ((PMID: 40476345). Nevertheless, I think the added references (8,9) as suggested by the reviewer, and the following paragraph smooth things nicely:
"In vitro studies using cell cultures and brain slices add support to these conclusions, as they demonstrate utilization of lactate for neuronal function, glycolytic glucose utilization by astrocytes and lactate transport to neurons [47-53]. Several in vivo studies also show lactate utilization by the human brain by injecting hyperpolarized 13C-lactate, which is immediately converted to 13C-pyruvate and oxidized, a conversion that occurs most likely by mitochondrial lactate dehydrogenase [54]. Moreover, lactate appears to be attracted to the brain’s gray matter more than to its white matter [55]."
I agree with the reviewer thar the manuscript would also serve as a stronger review - especially for a broader readership - if it addressed the many shortcomings of the BOLD methodology in greater detail. Hence, the following paragraph and references were added:
"Finally, since the BOLD fMRI technique has become the leading methodology in a multitude of studies investigating cerebral energy metabolism in rest and during activity, in health and disease, it is important to consider not only its promises, but some of its pitfalls and drawbacks [65-67]. To this list of the drawbacks, one should add its inability to detect extravascular oxygen, oxygen that active neural tissue utilizes for mitochondrial ATP production."
Reviewer 3 Report
Comments and Suggestions for Authors
In this review the author addresses the long-standing debate on aerobic vs. anaerobic glycolysis. Researchers working on brain metabolism have still not been able to answer how neurons can adapt their glucose metabolism to meet their diverse metabolic needs.
Comment 1: Several studies have suggested that oxidative phosphorylation is majorly utilized at rest while glycolysis is preferred during intense activity (Ashrafi et al, 2017, Meyong et al, 2024) because it is a lot faster mechanism than oxidative phosphorylation. This phenomenon can explain an increase in lactate production but not oxygen consumption upon prolonged stimulation. One can argue that since all of the pyruvate generated by glycolysis is not fed into the TCA cycle during high activity period, therefore a major portion gets converted into lactate.
Comment 2: A recent article (Tiwari et al, 2024) shows that blocking the mitochondrial pyruvate carrier results in impaired synaptic vesicle recycling as well as presynaptic ATP levels. If lactate is the ultimate product of glycolysis, they would not have seen a reduction in presynaptic ATP levels or SV recycling when mitochondrial pyruvate transport is blocked.
Comment 3: Wei et al, 2023 have suggested compartmentalization of glycolysis and oxidative phosphorylation in neurons. According to this paper glycolysis is a preferred phenomenon in soma whereas synaptic terminals prefer oxidative phosphorylation. Another study (Pathak et al, 2015) demonstrates that even though 30-50% of synaptic terminals are devoid of mitochondria yet ATP measurements across synaptic terminals suggest that ATP gets diffused from the terminals with mitochondrial to the terminals devoid of mitochondria to support neurotransmission. Therefore, it would be interesting if the author could share his views on how compartmentalization of glycolysis and oxidative phosphorylation can work in meeting diverse metabolic demand of neurons during different physiological/developmental stages.
Comment 4: Since the author is an expert in this field it would be enlightening to know their views on how an in-depth in vivo study can be designed to investigate the metabolic regulation of synaptic transmission.
Author Response
My thanks to Reviewer 3 for the excellent comments and suggestions. I was hesitating whether to include my responses to the reviewer's comments in the manuscript, or to respond to them here without including them. After some consideration, I decided that the manuscript and its readers could benefit from their inclusion. I posted my responses to the first three comments below, as they appear in the revised manuscript. I believe that the reviewer's points do improve the manuscript.
Comment 1: Several studies have suggested that oxidative phosphorylation is majorly utilized at rest while glycolysis is preferred during intense activity (Ashrafi et al, 2017, Meyong et al, 2024) because it is a lot faster mechanism than oxidative phosphorylation. This phenomenon can explain an increase in lactate production but not oxygen consumption upon prolonged stimulation. One can argue that since all of the pyruvate generated by glycolysis is not fed into the TCA cycle during high activity period, therefore a major portion gets converted into lactate.
Comment 2: A recent article (Tiwari et al, 2024) shows that blocking the mitochondrial pyruvate carrier results in impaired synaptic vesicle recycling as well as presynaptic ATP levels. If lactate is the ultimate product of glycolysis, they would not have seen a reduction in presynaptic ATP levels or SV recycling when mitochondrial pyruvate transport is blocked.
Comment 3: Wei et al, 2023 have suggested compartmentalization of glycolysis and oxidative phosphorylation in neurons. According to this paper glycolysis is a preferred phenomenon in soma whereas synaptic terminals prefer oxidative phosphorylation. Another study (Pathak et al, 2015) demonstrates that even though 30-50% of synaptic terminals are devoid of mitochondria yet ATP measurements across synaptic terminals suggest that ATP gets diffused from the terminals with mitochondrial to the terminals devoid of mitochondria to support neurotransmission. Therefore, it would be interesting if the author could share his views on how compartmentalization of glycolysis and oxidative phosphorylation can work in meeting diverse metabolic demand of neurons during different physiological/developmental stages.
"several recent studies either support the non-oxidative glucose consumption as the metabolic process that provide ATP to active neurons, or that lactate is not the oxidative mitochondrial substrate of OXPHO. An in vitro study determined that lactate transport from astrocytes via monocarboxylate transporter 1 (MCT1) to neurons via MCT2 is not necessary to maintain synchronized synaptic transmission [57], and a recent editorial highlights these findings to argue for “the absolute necessity of neuronal glucose metabolism to maintain brain function.” [58]. Another study shows that transport of pyruvate by the mitochondrial pyruvate carrier (MPC) regulates presynaptic metabolism and neurotransmission [59]. This finding does not necessarily indicate that lactate is not the final glycolytic end-product and the oxidative mitochondrial substrate. The MPC is located on the inner mitochondrial membrane [60]. However, the mitochondrial MCT is located on the outer membrane and is responsible for lactate transport into the mitochondrion, where it is oxidized by mitochondrial lactate dehydrogenase (mLDH) to pyruvate, which is then transported by the MPC to be converted to acetyl Co-enzyme A (acetyl CoA [61-63]. Wei et al. [64] suggested a compartmentalization of neuronal energy metabolism, where OXPHOS takes place mainly in axonal terminals, while glycolysis is the main supplier of ATP in neuronal somata. The investigators suggest that their findings support the concept of non-oxidative glucose consumption as the principal metabolic ATP producer during neuronal activity. They also found that the glycolytic enzyme pyruvate kinase 2 (PKM2) is more prevalent in the neuronal somata than in the terminals. Deletion of the gene pmk2 in mice caused neuronal somata to switch from glycolysis to OXPHOS, which increased oxidative damage. The role of glycolysis in protecting neurons against oxidative damage has been documented [51], where the protection is provided by the glycolytic end-product, lactate, not by pyruvate. Therefore, it is worth re-emphasizing that the glycolytic pathway always ends with lactate, the oxidative mitochondrial substrate, and an effective neuroprotectant against oxidative damage. As investigators continue to promote the dogma that glycolysis ends with pyruvate (in the presence of oxygen), the needless debate over which metabolic process fuels the active brain will continue to divide the research field of brain energy metabolism." The bold font here, does not appear in the manuscript text. I chose to use it here to emphasize my strong belief that at the root of many of the disagreements among brain energy metabolism researchers is the wrong concept that, somehow, glycolysis ends with pyruvate, not with lactate. This old, dogmatic concept should not continue to be used, not in research and not in the classroom.
Comment 4: Since the author is an expert in this field it would be enlightening to know their views on how an in-depth in vivo study can be designed to investigate the metabolic regulation of synaptic transmission.
I appreciate the reviewer's trust and request, but I think such elaboration is outside the scope of this manuscript. Of course, it would be something I am happy to discuss with the reviewer as a possible future experimental investigation.
Round 2
Reviewer 2 Report
Comments and Suggestions for Authors
All concerns/suggestions have been addressed.
Reviewer 3 Report
Comments and Suggestions for Authors
The author has responded adequately.